# A multicenter prospective study to determine the optimal range of lymph node dissection in pancreatic cancer surgery after neoadjuvant chemotherapy (LYMRIN-Trial): Project study by the Japan Pancreas Society and JON 2302-P

Nana Kimura[1], Yoshihiro Shirai[1], Ayaka Itoh[1], Katsuhisa Hirano[1], Kazuto Shibuya[1], Kenta Murotani[2,3], Makoto Ueno[4], Sohei Satoi[5], Hiroaki Nagano[6], Junji Furuse[4], Yoshifumi Takeyama[7], Tsutomu Fujii[1]*, LYMRIN-Trial collaborators[¶]

1 Department of Surgery and Science, Faculty of Medicine, Academic Assembly, University of Toyama, Toyama, Japan, 2 Biostatistics Center, Kurume University Graduate School of Medicine, Kurume, Fukuoka, Japan, 3 School of Medical Technology, Kurume University, Kurume-shi, Fukuoka, Japan, 4 Department of Gastroenterology, Kanagawa Cancer Center, Yokohama, Kanagawa, Japan, 5 Department of Surgery, Kansai Medical University, Osaka, Japan, 6 Department of Gastroenterological, Breast and Endocrine Surgery, Yamaguchi University Graduate School of Medicine, Yamaguchi, Japan, 7 Department of Surgery, Kindai University Faculty of Medicine, Osaka, Japan.

¶ LYMRIN-Trial collaborators is provided in the Acknowledgments.
☯ These authors contributed equally to this work.
* fjt@med.u-toyama.ac.jp

## Abstract

### Introduction

Clear evidence regarding the optimal extent of lymph node dissection during pancreatectomy for pancreatic ductal adenocarcinoma (PDAC) is lacking. The index of estimated benefit of lymph node dissection is useful for evaluating the effectiveness of dissection at each lymph node station, but a prospective study is needed for accurate assessment. The aim of the LYMRIN trial is to determine the optimal station of lymph node dissection in pancreatectomy after neoadjuvant chemotherapy using gemcitabine and S-1 (GS) by evaluating the rate of lymph node metastasis and the index of estimated benefit of each type of lymph node dissection.

### Methods

This multicenter prospective interventional trial in Japan that will include a total of 545 PDAC patients (head: 200 patients, body: 165 patients, and tail 180 patients) from 42 leading institutions and hospitals in Japan. Patients diagnosed with resectable PDAC after neoadjuvant GS therapy will be considered eligible for inclusion. Eligible patients will undergo pancreatoduodenectomy or distal pancreatectomy with regional lymph node dissection. The primary endpoint is the percentage of metastases in each lymph node station.

**Data availability statement:** No datasets were generated or analysed during the current study. All relevant data from this study will be made available upon study completion.

**Funding:** This study was funded in part by the Committee for the Promotion of Clinical Research of the Japan Pancreas Society (Grant number 2022-01). The funders had no role in study design, data collection and analysis, decision to publish, or preparation of the manuscript.

**Competing interests:** The authors have declared that no competing interests exist.

**List of abbreviations:** PDAC, pancreatic ductal adenocarcinoma, GS, gemcitabine and S-1, RCT, randomized controlled trial, NAC, neoadjuvant chemotherapy, SMV, superior mesenteric vein, PV, portal vein, CHA, common hepatic artery, SMA, superior mesenteric artery

## Discussion

The results of this study will clarify the optimal extent of lymph node dissection, allowing for a more conservative surgical approach, thereby avoiding unnecessary invasive procedures for patients.

## Trial registration

University Hospital Medical Information Network Clinical Trials Registry, UMIN 000051879. Registered 1 October 2023, https://center6.umin.ac.jp/cgi-open-bin/ctr/ctr_view.cgi?recptno=R000059119

## Introduction

Pancreatic ductal adenocarcinoma (PDAC) is one of the most lethal types of gastrointestinal cancers and the seventh leading cause of cancer-related death worldwide [1]. The number of deaths due to pancreatic cancer continues to increase, with a 5-year survival rate of only 12% [2]. Pancreatic cancer can be classified as resectable, borderline resectable, or unresectable based on whether R0 surgery without gross or histologic evidence of residual cancer is possible by standard surgery [3]. According to the results of the Prep-02/JSAP-05 study, the use of gemcitabine plus S-1 (GS) therapy as a neoadjuvant chemotherapy (NAC) significantly prolonged overall survival [4,5]. GS therapy is recommended as NAC for treating resectable pancreatic cancer according to the 2022 Clinical Practice Guidelines for Pancreatic Cancer in Japan [6].

In the 1980s and 1990s, radical cure was prioritized for surgical resection, and extended lymph node and plexus dissection were performed for resectable pancreatic cancer. However, several randomized controlled trials (RCTs) [7,8] showed that extended lymph node dissection in pancreatic cancer patients did not improve prognosis and these findings led to the recommendation against performing extended lymph node dissection [6]. While extended lymph node dissection has been ruled out for treatment, the optimal range of standard lymph node dissection in the head, body or tail of the pancreas has not been reported on. Moreover, the latest guidelines from the National Comprehensive Cancer Network and Japan Pancreas Society do not specify the appropriate station for lymph node dissection.

The index of estimated benefit of lymph node dissection [9] is calculated as the metastasis rate × survival rate of metastatic patients, which is useful in deciding whether to perform lymph node dissection. This index is used to predict the effect of each lymph node dissection in patients with gastric cancer [9]; however, it cannot be accurately calculated when there is variation in the area and precision of dissection, so retrospective studies are unable to accurately assess the benefit of dissection. The aim of this prospective trial is to determine the optimal station to perform lymph node dissection during pancreatectomy for PDAC patients after NAC. This will be achieved by evaluating the rate of lymph node metastasis and the index of estimated benefit of each type of lymph node dissection.

## Methods/design

### Aim

The aim of the LYMRIN trial is to determine the optimal range of lymph node dissection for patients with resectable pancreatic cancer after NAC. This will be achieved by clarifying the index of estimated benefit of lymph node dissection, which will be calculated by the metastasis rate of each lymph node station and the survival rate of the patient with lymph node metastasis. Clinical study protocol was described in the S1 Protocol.

### Study design and setting

The LYMRIN trial is designed as a multicenter prospective interventional trial that will take place in Japan and will include a total of 545 PDAC patients (head: 200 patients, body: 165 patients, and tail 180 patients) from 42 leading institutions and hospitals in Japan (S1 File). Institutions participating in the study are required to be an HPB board certified and perform at least 40 pancreatectomies per year. The recruitment period for this study is from the date of approval to perform the study at each institution to September 30, 2026. Data collection and follow-up period is 5 years after the end of recruitment. The total study period, including results, is until September 30, 2031.

### Endpoints

This clinical trial will primarily evaluate the metastatic rate of each lymph node station in a standard cohort treated with NAC. The secondary endpoints include the index of estimated benefit of each lymph node dissection (metastatic rate x 3- or 5-year survival), disease-free survival, overall survival, cancer-specific survival, postoperative complications, the location of the relapse site in the standard and nonstandard cohorts, and the metastatic rate of each lymph node station in the nonstandard cohort. The results will be published in a conference or in a paper.

### Study population and eligibility criteria

The subjects of this study will be patients with PDAC whose resectability classification is judged to be resectable according to the 8th edition of the Japanese Classification of Pancreatic Carcinoma edited by the Japan Pancreas Society [10] after NAC. The detailed eligibility criteria are described in Table 1.

### Registration

Fig 1 shows a flow diagram of the LYMRIN trial. Patients who intend to receive GS combination therapy will be enrolled in the standard NAC cohort if they have completed two or less courses of preoperative GS therapy and have undergone imaging and blood collection tests to evaluate efficacy. Gemcitabine will be administered at a dose of 1,000 mg/m$^2$ on days 1 and 8, and oral S-1 will be administered at a dose of 40 mg/m$^2$ twice daily on days 1–14 [11]. If the patient has no new lesions or distant metastases and is deemed eligible for radical primary tumor resection, the patient will be enrolled in the standard NAC cohort. All potential trial participants will be asked to provide informed consent or assent by the principal investigator or co-investigators. All participants will be required to provide informed consent before enrolling in this trial. Patients will be enrolled within 28 days of the last administration of NAC. Patients who have received other preoperative chemotherapy (gemcitabine plus nab-paclitaxel, FOLFIRINOX (original or modified), or ≥3 courses of GS) will be enrolled into the nonstandard NAC cohort after a similar evaluation. The participant timeline is shown in Fig 2.

### Sample size estimation

The expected enrollment for this trial is 545 patients in the standard NAC cohort (200 patients with pancreatic head cancer, 165 patients with pancreatic body cancer, and 180 patients with pancreatic tail cancer). The number of patients,

**Table 1. Eligibility criteria.**

| Inclusion criteria | |
|---|---|
| (1) | Histological diagnosis: pancreatic ductal adenocarcinoma |
| (2) | The primary tumor is diagnosed as resectable |
| (3) | No history of upper abdominal surgery |
| (4) | Surgical resection with regional lymph nodes (Table 2) dissection is possible with imaging |
| (5) | Peritoneal washing cytology: negative (if applicable) |
| (6) | Neoadjuvant chemotherapy was performed |
| (7) | Tumors determined to be suitable for radical primary resection within 28 days after neoadjuvant chemotherapy |
| (8) | Planned pancreatoduodenectomy for pancreatic head cancer, distal pancreatectomy for pancreatic body-tail cancer |
| (9) | Age: 18 years or older |
| (10) | Performance status (Eastern Cooperative Oncology Group scale): 0–1 |
| (11) | Ability to understand and willingness to sign written informed consent document |
| **Exclusion criteria** | |
| (1) | Active multiple primary cancers (synchronous or asynchronous within 5 years) |
| (2) | Serious comorbidities (heart failure, interstitial pneumonia, renal failure, liver failure, intestinal paralysis, intestinal obstruction, poorly controlled diabetes, poorly controlled hypertension, etc.) |
| (3) | Protocol treatment (radical primary tumor resection and regional lymph node dissection) cannot be safely performed |
| (4) | Receiving pre-treatment (radiotherapy, immunotherapy, etc.) other than neoadjuvant chemotherapy |
| (5) | Pregnant or lactating women and women of childbearing potential |
| (6) | Severe psychological or neurological disease |

the lymph nodes (pancreatic head: no. 14, pancreatic body: no. 10, and pancreatic tail: no. 8a) will be selected because they are the most controversial in terms of dissection or not selected due to the low frequency of lymph node metastasis. The metastasis rate of each patient will be used as the basis for calculation. The Pancreatic Cancer Registry Database in the National Clinical Database showed that the percentage of pathologically positive lymph node No. 14 for pancreatic head cancer is 10.9%. The Wilson score method was used to calculate the number of patients to be enrolled to achieve a probability of 90% or greater that the metastatic rate would fall within the 95% confidence interval of ±5%. The number of enrolled patients was estimated to be 183. Considering that approximately 10% of patients may be excluded from the analysis due to ineligibility, the target enrollment number in the No. 14 lymph node is set at 200 patients. Similarly, for patients with pancreatic body cancer, we estimated 151 patients will be required with a metastatic rate of 3.5% in the No. 10 lymph node and a 95% confidence interval width of at least 80% for the probability of being within 3.5%. Considering that approximately 10% of patients will be excluded from the analysis, the target enrollment number is set at 165 patients. For patients with pancreatic tail cancer, the percentage of pathologically positive No. 8a lymph nodes is 3.2%. The number of cases required to achieve a probability of 80% or greater within the 95% confidence interval of ±3.2% is estimated to be 166. Therefore, the target number of enrollment is set at 180 patients.

## Statistical analysis plan

All the statistical analyses will be performed using the full analysis set (FAS). The FAS is defined as the population of patients who achieved R0 resection after completion of the study protocol, excluding those who did not meet the inclusion criteria or who met the exclusion criteria. The percentages of metastases in lymph nodes no.14 (along the superior mesenteric artery), no. 10 (at the splenic hilum) and no. 8a (in the anterosuperior group along the common hepatic artery),

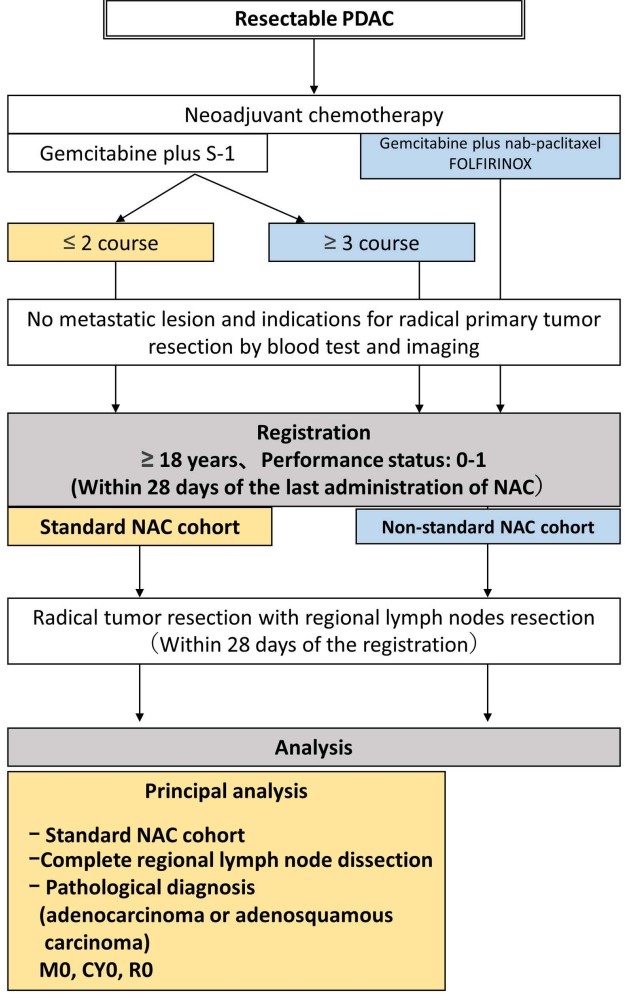

**Fig 1. Flow diagram.** Flow diagram of the LYMRIN trial.

which are the primary endpoints, will be determined and the 95% confidence intervals will be estimated using the Clopper–Pearson method. Kaplan-Meier curves will be used to represent time-to-event variables.

## Interventions

**Treatment protocol.** The treatment protocol used in this study involves radical primary pancreatectomy (pancreatoduodenectomy or distal pancreatectomy) and regional lymph node dissection from the start to the end of surgery. The border between the pancreatic head and body is defined as the left side of the superior mesenteric vein (SMV) and portal vein (PV). The neck of the pancreas (a region anterior to the SMV and PV) and uncinate process are included in the pancreatic head. The border between the pancreatic body and tail is defined as the left border of the abdominal aorta (Fig 3[12]). The defined procedure is pancreatoduodenectomy for pancreatic head cancer and distal pancreatectomy for pancreatic body and tail cancer. In this study, patients who undergo pancreatoduodenectomy for pancreatic body cancer will be excluded. Patients who undergo total pancreatectomy or combined resection of the major arteries (celiac artery, hepatic artery, and superior mesenteric artery) will also be also excluded from the analysis. Surgical resection in this study will allow open surgery, laparoscopic pancreatectomy, and robot-assisted pancreatectomy. This study includes several

| | STUDY PERIOD | | | | | | | | |
|---|---|---|---|---|---|---|---|---|---|
| | Enrolment | | Registration | Post-registration | | | | | |
| TIMEPOINT | -28 days | -14 days | 0 | Operation | Pathological findings | POD 30 | 1 Month-3 years | 3-5 years | After 5 years |
| ENROLMENT: | | | | | | | | | |
| Eligibility screen | | | X | | | | | | |
| Informed consent | | | X | | | | | | |
| Blood test | | X | | | | X | X | X | X |
| Tumor markers | | X | | | | | X | X | X |
| Enhanced CT | X | | | | | | X | X | X |
| Peritoneal cytology | | | X | X | | | | | |
| Registration | | | X | | | | | | |
| INTERVENTION: | | | | | | | | | |
| Operation | | | | X | | | | | |
| ASSESSMENT: | | | | | | | | | |
| Complications | | | | X | | X | X | X | X |
| Pathological findings | | | | | X | | | | |
| Survivals | | | | | | | X | X | X |

**Fig 2. Timeline.** Participant timeline of the LYMRIN trial.

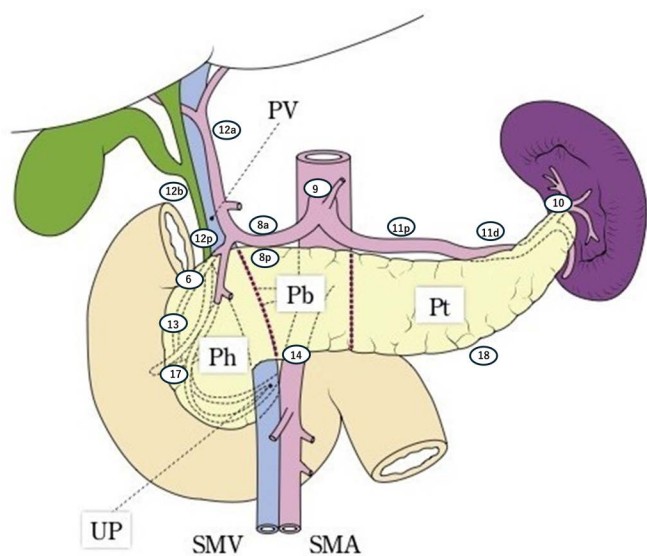

**Fig 3. Borders and lesional lymph nodes.** The border of the pancreas head, body, and tail. Lesional lymph nodes. Lymph node 14 was classified as 14t on the tumor side or 14op on the tumor opposite side. Ph: head of the pancreas, Pb: body of the pancreas, Pt: tail of the pancreas, PV: portal vein, SMA: superior mesenteric artery, SMV: superior mesenteric vein, UP: uncinate process.

requirements with additional interventions. In the pancreas head tumor, No. 14 op lymph node (Lymph nodes along the superior mesenteric artery; opposite side of tumor) is not a regional lymph node in Japanese classification of pancreatic carcinoma by the Japan Pancreas Society: Eighth edition [10], but our protocol needs to dissect No. 14op lymph nodes with pancreatoduodenectomy. The criterion for discontinuing the intervention for a trial participant is that the treatment protocol is deemed medically unsafe to carry out by the principal investigator or co-investigators. The principal investigators and co-investigators will be notified of important changes to the protocol at each site via email.

**Regional lymph nodes dissection.** Station numbers and names of lymph nodes related to the pancreas are defined by Japanese classification of pancreatic carcinoma by the Japan Pancreas Society: Eighth edition [10] (S1 Table). The surgeon will perform the designated lymph nodes dissection based on the tumor location (Table 2, S1 Table) (Fig 3[12]). Pancreatoduodenectomy for pancreatic head tumors requires dissection of 14op lymph node, which is outside the regional lymph node. This trial does not prescribe how to divide and determine lymph node stations.

**Postoperative follow-up.** Adjuvant chemotherapy is recommended as standard therapy with 6 months of S-1 therapy, but the indication, regimen, duration, and dosage of adjuvant therapy may be determined on a patient-by-patient basis. Follow-up will be conducted for 5 years after the completion of enrollment. Blood tests, tumor markers, contrast-enhanced CT, and late complications will be evaluated every 3 months through the third postoperative year. Recurrence is determined clinically using imaging and tumor markers. The site of recurrence and cause of death will be recorded.

## Ethics approval and consent to participate

All the investigators involved in this research will conduct this study according to the Helsinki Declaration and the Ethical Guidelines for Clinical Studies of the Ministry of Health, Labor and Welfare of Japan. The protocol was approved by the Institutional Review Board (IRB) of the University of Toyama (No. R2023163) and all participating institutions. This trial is registered on the UMIN Clinical Trials Registry (UMIN 000051879). All subjects for this study will be provided a consent form describing this study and providing sufficient information for subjects to make an informed decision about their participation in this study. The consent form will be submitted with the protocol for review and approval by the IRB for the study.

## Data collection and management

Participants will be completely anonymized, and data will be managed by registration numbers. A correspondence table will be created to link registration numbers to participants' personal information so that anonymized subjects can be matched when necessary. The electronic data will be stored on a secure cloud server. Only investigators authorized by the principal investigator will have access to the data, and a list of those with access privileges will be created.

## Monitoring

Central monitoring will be performed by an independent data monitoring committee. The purpose of monitoring is to ensure that the human rights and welfare of human subjects are protected in the context of clinical research, that research data are accurate, complete, and verifiable against source documents, and that clinical research is conducted in accordance with the protocol and applicable regulatory requirements. The monitoring committee reports serious adverse events to the efficacy and safety assessment committee.

## Efficacy and safety assessment

The trial will be monitored by the efficacy and safety monitoring committee. The principal investigator will ask the committee to review the opinions about adverse events and the suitability of responses and other actions. The committee will review and consider the content of the report and make recommendations to the investigator regarding the next steps.

**Table 2. Lymph nodes to be dissected by tumor location.**

| Tumor location of pancreas | Lymph nodes |
|---|---|
| Head | 6, 8a, 8p, 12a, 12b, 12p, 13, 14t,14op, 17 |
| Body | 8a, 8p, 9, 10, 11p, 11d, 14t, 18 |
| Tail | 8a. 9, 10, 11p, 11d, 18 |

## Discussion

Pancreatic cancer is a malignancy that is associated with a poor prognosis, and for which a curative treatment strategy must include surgery. The rate of lymph node metastasis in pancreatic cancer patients is more than 50% [10]. In the case of the pancreatic head, lymph node metastasis is observed in the peripancreatic head, common hepatic artery (CHA), superior mesenteric artery (SMA), and hepatoduodenal mesentery region, and in the case of the pancreatic body and tail, in the peripancreatic, spleen, and SMA regions. These lymph nodes were defined as the area of pancreatic cancer, and extended prophylactic lymph node dissection has been performed without metastasis. Recently, RCTs [7,8] have shown that extended lymph node dissection in pancreatic cancer patients does not improve prognosis, leading to the recommendation against performing extended lymph node dissection [6]. However, the results of these RCTs must be interpreted in the context of several limitations. First, the definition of an extended lymph node varies from study to study. Indeed, lymph nodes around the CHA, SMA and hepatoduodenal mesentery were assigned to either the standard or the expanded lymph node dissection group, depending on the study. Second, it has been verified that extended lymph node dissection is not necessary; however, the lymph nodes that should be dissected have not been evaluated. Moreover, the effect of the dissection of each lymph node station is unclear. Third, all studies were limited to the head of the pancreas and, therefore, whether these results are applicable to the body and tail of the pancreas remains unclear. Finally, the effect of neoadjuvant chemotherapy was not considered in these RCTs. The use of perioperative multidisciplinary management and post recurrence chemotherapy has evolved significantly since the design of the previously reported trials. Thus, the role of surgery needs to be reconsidered when considering treatment strategies for pancreatic cancer.

Several retrospective studies have reported that the index of estimated benefit of lymph node dissection is useful for determining the area of lymph node dissection, [9]. In pancreatic body and tail cancer patients, dissection around the splenic artery and inferior pancreatic lymph nodes is reported to be more effective, while dissection around the left gastric artery and common hepatic artery is less effective [13]. In addition, several analyses of the positivity rate and prognosis by lymph node station have been reported [14,15]. However, a prospective study which accounts for the accuracy of the extent of lymph node dissection is needed to evaluate the exact effect of dissection. For accurate analysis and evaluation, this study requires participating institutions to perform more than 40 pancreatectomies per year and to involve a Japanese Pancreatic Society council member in the study to ensure surgery quality. In addition, the extent of lymph node dissection will be clearly defined, and only patients who will be able to complete the NAC protocol will be allowed to enroll.

This study will accumulate data on cases by the following regions: pancreatic head, pancreatic body, and pancreatic tail. Pancreatic body tumors are often resected by pancreatoduodenectomy or distal pancreatectomy. Only distal pancreatectomies will be included in this study, and pancreatoduodenectomies and total pancreatectomies will be excluded from the analysis. There are few reports about pancreatic body tumors because the number of tumors in the body of the pancreas is relatively small compared to that in the head of the pancreas. The results of this study will be valuable as prognostic data by localization, especially for the pancreatic body.

The data obtained from a single regimen will be reliable and meaningful. GS therapy, such as NAC, is widely used as the standard treatment in Japan [6]. NAC was suggested to reduce lymph node metastasis rates, which is an interesting finding that contrasts previous reports. In addition, data from patients treated with preoperative regimens such as gemcitabine plus nab-paclitaxel or FOLFIRINOX will be collected and analyzed as a nonstandard cohort. In doing so, positive lymph node rates in response to neoadjuvant treatment regimens can be evaluated.

Although NAC-GS therapy and surgical resection with regional lymph nodes resection are standard treatments for resectable PDAC, the study was considered an interventional trial because the possibility of additional invasiveness cannot be excluded when performing definitive lymph node dissection or contralateral lymph node dissection of the SMA in the pancreatic head.

The results of this study will clarify the optimal extent of lymph node dissection, allowing for a more conservative surgical approach and avoiding unnecessary invasive procedures for the patient. A reduction in invasiveness leads to a lower rate of complications, which may provide patient benefits such as early initiation of adjuvant chemotherapy and improved prognosis.

## Supporting information

**S1 Table. Station numbers and names of lymph nodes related to the pancreas.** Station numbers and names of lymph nodes related to the pancreas in this study.
(DOCX)

**S1 File. Institution list.** List of facilities participating in this study.
(DOCX)

**S1 Protocol. Study protocol.** Study protocol of LYMRIN trial.
(DOCX)

**S2 File. SPRIT check list.**
(DOCX)

## Acknowledgments

We appreciate the following members of LYMRIN-Trial: Project study by the Japan Pancreas Society and JON 2302-P for their valuable support in data collection and drafting of the manuscript.;Hideki Yokoo, Division of Hepato-Biliary-Pancreatic and Transplantation Surgery, Asahikawa Medical University, Hokkaido, Japan.; Sadatoshi Shimizu, Department of Hepato-Biliary-Pancreatic Surgery, Osaka City General Hospital, Osaka, Japan.; Hidetoshi Eguchi, Department of Gastroenterological Surgery, Graduate School of Medicine, Osaka University, Osaka, Japan.; Toshihiko Masui, Department of Surgery Kurashiki Central Hospital, Okayama, Japan.; Keiichi Okano, Department of Gastroenterological Surgery, Faculty of Medicine, Kagawa University, Kagawa, Japan.; Takao Ohtsuka, Department of Digestive Surgery, Graduate School of Medical and Dental Sciences, Kagoshima University, Kagoshima, Japan.; Masafumi Nakamura, Department of Surgery and Oncology, Graduate School of Medical Sciences, Kyushu University, Fukuoka, Japan.; Etsuro Hatano, Department of Surgery, Graduate School of Medicine, Kyoto University, Kyoto, Japan.; Yoshihiro Sakamoto, Department of Hepato-Biliary-Pancreatic Surgery, Kyorin University Hospital, Tokyo, Japan.; Ippei Matsumoto, Department of Surgery, Faculty of Medicine, Kindai University, Osaka, Japan.; Minoru Kitago, Department of Surgery, Keio University, Tokyo, Japan.; Amane Takahashi, Department of Gastroenterological Surgery, Saitama Cancer Center, Saitama, Japan.; Masafumi Imamura, Department of Surgery, Surgical Oncology and Science, Sapporo Medical University School of Medicine, Hokkaido, Japan.; Takeshi Aoki, Division of General and Gastroenterological Surgery, Department of Surgery, School of Medicine, Showa University, Tokyo, Japan.; Naohiro Sata, Division of Gastroenterological, General and Transplant Surgery, Department of Surgery, Jichi Medical University, Tochigi, Japan.; Toshiki Rikiyama, Department of Surgery, Saitama Medical Center, Jichi Medical University, Saitama, Japan.; Akio Saiura, Department of Hepatobiliary-Pancreatic Surgery, Juntendo University Graduate School of Medicine, Tokyo, Japan.; Atsushi Kato, Department of Hepato-Biliary-Pancreatic Surgery, Chiba Cancer Centre, Chiba, Japan.; Masayuki Ohtsuka, Department of General Surgery, Graduate School of Medicine, Chiba University, Chiba, Japan.; Hirofumi Akita, Department of Gastroenterological Surgery, Osaka International Cancer Institute, Osaka, Japan.; Naoto Yamamoto, Department of Gastrointestinal Surgery, Kanagawa Cancer Center, Kanagawa, Japan.; Yuichi Nagakawa, Department of Gastrointestinal and Pediatric Surgery, Tokyo Medical University, Tokyo, Japan.; Michiaki Unno, Department of Surgery, Tohoku University Graduate School of Medicine, Miyagi, Japan.; Hideyuki Yoshitomi, Department of Surgery, Dokkyo Medical University Saitama Medical Center, Saitama, Japan.; Taku

Aoki, Department of Hepato-Biliary Pancreatic Surgery, Dokkyo Medical University, Tochigi, Japan.; Akimasa Nakao, Department of Gastroenterological Surgery, Nagoya Central Hospital, Aichi, Japan.; Hideki Takami, Department of Gastroenterological Surgery, Nagoya University Graduate School of Medicine, Aichi, Japan.; Masayuki Sho, Department of Surgery, Nara Medical University, Nara, Japan.; Seiko Hirono, Department of Gastroenterological Surgery, Hyogo Medical University, Hyogo, Japan.; Kenichi Hakamada, Department of Gastroenterological Surgery, Hirosaki University Graduate School of Medicine, Aomori, Japan.; Akihiko Oshita, Department of Surgery, Onomichi General Hospital, Hiroshima, Japan; Kenichiro Uemura, Department of Surgery, Graduate School of Biomedical and Health Sciences, Hiroshima University, Hiroshima, Japan.; Akihiko Horiguchi, Department of Gastroenterological Surgery, Fujita Health University School of Medicine Bantane Hospital, Aichi, Japan.; Takeshi Takahara, Department of Surgery, Fujita Health University, Aichi, Japan.; Satoshi Hirano, Department of Gastroenterological Surgery II, Faculty of Medicine, Hokkaido University, Hokkaido, Japan.; Shugo Mizuno, Department of Hepatobiliary Pancreatic and Transplant Surgery, Mie University Graduate School of Medicine, Mie, Japan.; Atsushi Nanashima, Department of Surgery, Faculty of Medicine, University of Miyazaki, Miyazaki, Japan.; Fuyuhiko Motoi, Department of Surgery I, Yamagata University Faculty of Medicine, Yamagata, Japan.; Manabu Kawai, Second Department of Surgery, School of Medicine, Wakayama Medical University, Wakayama, Japan.

## Trial status

Enrollment for the LYMRIN trial began in October 2023. At the time of submission for this paper (Nov 2024), the protocol version is ver. 1.3. The completion date is estimated to be September 2031.

## Author contributions

**Conceptualization:** Katsuhisa Hirano, Makoto Ueno, Sohei Satoi, Tsutomu Fujii.

**Formal analysis:** Kenta Murotani.

**Funding acquisition:** Tsutomu Fujii.

**Methodology:** Kenta Murotani.

**Project administration:** Tsutomu Fujii.

**Supervision:** Makoto Ueno, Sohei Satoi, Hiroaki Nagano, Junji Furuse, Yoshifumi Takeyama, Tsutomu Fujii.

**Writing – original draft:** Nana Kimura, Yoshihiro Shirai, Ayaka Itoh, Katsuhisa Hirano, Kenta Murotani, Tsutomu Fujii.

**Writing – review & editing:** Yoshihiro Shirai, Kazuto Shibuya, Makoto Ueno, Sohei Satoi, Hiroaki Nagano, Junji Furuse, Yoshifumi Takeyama, Tsutomu Fujii.

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
