## [Decision Letter · Decision Letter 0]

Feb 09 2025

Dear Dr. Fujii,

Thank you for submitting your manuscript to PLOS ONE. After careful consideration, we feel that it has merit but does not fully meet PLOS ONE’s publication criteria as it currently stands. Therefore, we invite you to submit a revised version of the manuscript that addresses the points raised during the review process.

We look forward to receiving your revised manuscript.

Kind regards,

Peng Zhang, Ph.D.

Academic Editor

PLOS ONE

Journal Requirements: When submitting your revision, we need you to address these additional requirements. 1. Please ensure that your manuscript meets PLOS ONE's style requirements, including those for file naming. The PLOS ONE style templates can be found at https://journals.plos.org/plosone/s/file?id=wjVg/PLOSOne_formatting_sample_main_body.pdf and https://journals.plos.org/plosone/s/file?id=ba62/PLOSOne_formatting_sample_title_authors_affiliations.pdf 2. We note that the grant information you provided in the ‘Funding Information’ and ‘Financial Disclosure’ sections do not match.  When you resubmit, please ensure that you provide the correct grant numbers for the awards you received for your study in the ‘Funding Information’ section. 3. Thank you for stating the following financial disclosure: "Committee for the Promotion of clinical research of the Japan Pancreas Society" Please state what role the funders took in the study.  If the funders had no role, please state: ""The funders had no role in study design, data collection and analysis, decision to publish, or preparation of the manuscript."" If this statement is not correct you must amend it as needed. Please include this amended Role of Funder statement in your cover letter; we will change the online submission form on your behalf. 4. Your ethics statement should only appear in the Methods section of your manuscript. If your ethics statement is written in any section besides the Methods, please move it to the Methods section and delete it from any other section. Please ensure that your ethics statement is included in your manuscript, as the ethics statement entered into the online submission form will not be published alongside your manuscript. 5. Please include a copy of Table 1 and 2 which you refer to in your text on page 8 and 11. 6. Please include your tables as part of your main manuscript and remove the individual files. Please note that supplementary tables (should remain/ be uploaded) as separate ""supporting information"" files 7. Please include captions for your Supporting Information files at the end of your manuscript, and update any in-text citations to match accordingly. Please see our Supporting Information guidelines for more information: http://journals.plos.org/plosone/s/supporting-information. 8. Please review your reference list to ensure that it is complete and correct. If you have cited papers that have been retracted, please include the rationale for doing so in the manuscript text, or remove these references and replace them with relevant current references. Any changes to the reference list should be mentioned in the rebuttal letter that accompanies your revised manuscript. If you need to cite a retracted article, indicate the article’s retracted status in the References list and also include a citation and full reference for the retraction notice.

Reviewers' comments:

Reviewer's Responses to Questions

**Comments to the Author**

1. Does the manuscript provide a valid rationale for the proposed study, with clearly identified and justified research questions?

Reviewer #1: Yes

Reviewer #2: Yes

2. Is the protocol technically sound and planned in a manner that will lead to a meaningful outcome and allow testing the stated hypotheses?

Reviewer #1: Yes

Reviewer #2: Yes

3. Is the methodology feasible and described in sufficient detail to allow the work to be replicable?

Reviewer #1: Yes

Reviewer #2: Yes

4. Have the authors described where all data underlying the findings will be made available when the study is complete?

Reviewer #1: Yes

Reviewer #2: Yes

5. Is the manuscript presented in an intelligible fashion and written in standard English?

Reviewer #1: Yes

Reviewer #2: Yes

You may also provide optional suggestions and comments to authors that they might find helpful in planning their study.

Reviewer #1: The manuscript is a study protocol for a trial examining incidences of lymph node metastases in certain lymph node stations according to tumor locations. It properly describes the follow of the study and is well-written. I have a few questions as below.

#1. This study seems prospective observational trial because the treatment included in the study is a standard treatment for resectable PDAC and participants will not receive any additional intervention.

#2. How are the numbers of lymph node stations determined? Are they determined by surgeons in the surgical fields and separately submitted, or by pathologists in the specimens?

Reviewer #2: Comments to PONE-D-24-55825

I would like to express my sincere gratitude for being appointed as a reviewer for this study protocol. This paper addresses a critical issue regarding the optimization of lymph node dissection in pancreatic cancer surgery, and I recognize its significant clinical relevance. To enhance its appeal and clarity for readers, I have the following suggestions.

Major comment

1. Since this is a multicenter study, a more detailed set of criteria for ensuring surgical consistency should be provided. In particular, the inclusion of HBP board certification as part of the facility qualifications needs to be explicitly addressed. Additionally, any related discussion currently in the manuscript should be moved from the Discussion section to the Methods section for clarity and coherence.

Minor comments

1. Line 332, The collaborator’s name is incorrect; it should be Masafumi, not Masashi (Masafumi Imamura, from Sapporo Medical University).

2. Lines 448, reference No. 10: This reference is a Japanese-language source, so please revise it to follow the appropriate citation format according to the journal's submission guidelines.

**Do you want your identity to be public for this peer review?** For information about this choice, including consent withdrawal, please see our Privacy Policy

Reviewer #1: No

Reviewer #2: **Yes: ** Yasutoshi Kimura

---

## [Author Response · Author response to Decision Letter 1]

6 Feb 2025

Response to the comments of the editor and the reviewers

Editor

Thank you for reviewing and handling our manuscript entitled " A multicenter prospective study to determine the optimal range of lymph node dissection in pancreatic cancer surgery after neoadjuvant chemotherapy (LYMRIN-Trial): Project study by the Japan Pancreas Society and JON 2302-P". We re-wrote the manuscript, tables, and supplementary files according to reviewers’ comments. We also responded to the Journal requirements one by one. The manuscript was brushed up. We have marked in red the changes made in the manuscript.

Journal Requirements:

1. [Please ensure that your manuscript meets PLOS ONE's style requirements, including those for file naming. The PLOS ONE style templates can be found at

https://journals.plos.org/plosone/s/file?id=wjVg/PLOSOne_formatting_sample_main_body.pdf and https://journals.plos.org/plosone/s/file?id=ba62/PLOSOne_formatting_sample_title_authors_affiliations.pdf]

We rewrote the manuscript in accordance with the PLOS ONE style and Study Protocol Article templates.

2. [We note that the grant information you provided in the ‘Funding Information’ and ‘Financial Disclosure’ sections do not match.]

As the editor said, we rewrote the funding information and matched ‘Funding Information’ and ‘Financial Disclosure’.

3. [Thank you for stating the following financial disclosure:

"Committee for the Promotion of clinical research of the Japan Pancreas Society"

Please state what role the funders took in the study. If the funders had no role, please state: ""The funders had no role in study design, data collection and analysis, decision to publish, or preparation of the manuscript."" If this statement is not correct you must amend it as needed. Please include this amended Role of Funder statement in your cover letter; we will change the online submission form on your behalf.]

We added the funder statement in the manuscript and cover letter as follows. The funder had no role in this study.

(Page3, line 37)

4. [Your ethics statement should only appear in the Methods section of your manuscript. If your ethics statement is written in any section besides the Methods, please move it to the Methods section and delete it from any other section. Please ensure that your ethics statement is included in your manuscript, as the ethics statement entered into the online submission form will not be published alongside your manuscript.]

We moved the ethics statement into the Methods section.

(Page 14, line 255)

5. [Please include a copy of Table 1 and 2 which you refer to in your text on page 8 and 11.]

We include a copy of Table 1 and 2 directly after the paragraph in which they are first cited.

6. [Please include your tables as part of your main manuscript and remove the individual files. Please note that supplementary tables (should remain/ be uploaded) as separate ""supporting information"" files]

We include Tables as part of your main manuscript and removed the individual files.

7. [Please include captions for your Supporting Information files at the end of your manuscript, and update any in-text citations to match accordingly. Please see our Supporting Information guidelines for more information: http://journals.plos.org/plosone/s/supporting-information.]

We added the captions of Supporting information at the end of the manuscript.

8. [Please review your reference list to ensure that it is complete and correct. If you have cited papers that have been retracted, please include the rationale for doing so in the manuscript text, or remove these references and replace them with relevant current references. Any changes to the reference list should be mentioned in the rebuttal letter that accompanies your revised manuscript. If you need to cite a retracted article, indicate the article’s retracted status in the References list and also include a citation and full reference for the retraction notice.]

We reviewed the reference list to ensure that it is complete and correct.

Reviewer 1:

Thank you for your important suggestion. We totally agree with your professional comments.

#1. [This study seems prospective observational trial because the treatment included in the study is a standard treatment for resectable PDAC and participants will not receive any additional intervention.]

We appreciate the professional comments. As the reviewer said, neoadjuvant GS therapy and surgical resection with dissection of regional lymph nodes are standard treatments for resectable PDAC. However, this study includes several requirements with additional interventions. In the pancreas head tumor, No. 14 op lymph node (Lymph nodes along the superior mesenteric artery; opposite side of tumor) is not a regional lymph node in General rules for the study of pancreatic cancer, 8th edition, but our protocol needs to dissect No. 14op lymph nodes with pancreatoduodenectomy. Although the treatment protocol does not deviate from the standard surgical procedures, the possibility of additional invasiveness cannot be ruled out due to achieving certain lymph nodes dissection. We determined that these additional invasions could occur as an intervention and developed a protocol. We added the comments in the discussion as follows.

‘Although NAC-GS therapy and surgical resection with regional lymph nodes resection are standard treatments for resectable PDAC, the study was considered an interventional trial because the possibility of additional invasiveness cannot be excluded when performing definitive lymph node dissection or contralateral lymph node dissection of the SMA in the pancreatic head.’

(Page 18, line 340)

2. [How are the numbers of lymph node stations determined? Are they determined by surgeons in the surgical fields and separately submitted, or by pathologists in the specimens?]

We appreciated the comment. This protocol does not specify how to determine lymph node stations. The methods of determination are assumed to be performed by the surgeon dividing in the surgical fields, or by surgeons or pathologists dividing in the specimens. We added the comments in Methods.

This trial does not prescribe how to divide and determine lymph node stations.

(Page 13, line 241)

Reviewer 2:

Thank you for your important suggestion. We totally agree with your professional comments.

Major comment

1. [Since this is a multicenter study, a more detailed set of criteria for ensuring surgical consistency should be provided. In particular, the inclusion of HBP board certification as part of the facility qualifications needs to be explicitly addressed. Additionally, any related discussion currently in the manuscript should be moved from the Discussion section to the Methods section for clarity and coherence.]

We appreciated the professional comments. As the reviewer said, Clarification of the requirements for participating institutions is important to ensure the credibility of the trial. Institutions participating in the study are required to be an HPB board certified and perform at least 40 pancreatectomies per year. We added the comment in the Methods.

‘Institutions participating in the study are required to be an HPB board certified and perform at least 40 pancreatectomies per year.’

(Page 8, line 133)

Minor comments

1. [Line 332, The collaborator’s name is incorrect; it should be Masafumi, not Masashi (Masafumi Imamura, from Sapporo Medical University).]

We appreciate the comment. We rewrote the manuscript.

2. [Lines 448, reference No. 10: This reference is a Japanese-language source, so please revise it to follow the appropriate citation format according to the journal's submission guidelines.]

We appreciated the comments. We changed the reference.

---

## [Decision Letter · Decision Letter 1]

Apr 11 2025

Dear Dr. Fujii,

Thank you for submitting your manuscript to PLOS ONE. After careful consideration, we feel that it has merit but does not fully meet PLOS ONE’s publication criteria as it currently stands. Therefore, we invite you to submit a revised version of the manuscript that addresses the points raised during the review process.

We look forward to receiving your revised manuscript.

Kind regards,

Peng Zhang, Ph.D.

Academic Editor

PLOS ONE

Journal Requirements:

Reviewers' comments:

Reviewer's Responses to Questions

**Comments to the Author**

1. Does the manuscript provide a valid rationale for the proposed study, with clearly identified and justified research questions?

Reviewer #1: Yes

Reviewer #2: Yes

2. Is the protocol technically sound and planned in a manner that will lead to a meaningful outcome and allow testing the stated hypotheses?

Reviewer #1: Yes

Reviewer #2: Yes

3. Is the methodology feasible and described in sufficient detail to allow the work to be replicable?

Reviewer #1: Yes

Reviewer #2: Yes

4. Have the authors described where all data underlying the findings will be made available when the study is complete?

Reviewer #1: Yes

Reviewer #2: Yes

5. Is the manuscript presented in an intelligible fashion and written in standard English?

Reviewer #1: Yes

Reviewer #2: Yes

You may also provide optional suggestions and comments to authors that they might find helpful in planning their study.

Reviewer #1: Although the manuscript has been well-revised, a few issues remain to be resolved.

#1. In the response, the authors noted “However, this study includes several requirements with additional interventions. In the pancreas head tumor, No. 14 op lymph node (Lymph nodes along the superior mesenteric artery; opposite side of tumor) is not a regional lymph node in General rules for the study of pancreatic cancer, 8th edition, but our protocol needs to dissect No. 14op lymph nodes with pancreatoduodenectomy.” If the protocol required additional interventions, they should be clearly stated in “Treatment protocol”. In addition, Table 2 included No. 14op in regional lymph nodes of the tumor in the pancreas head. As the authors noted No. 14op is not regional lymph nodes.

#2. Line 238-240, the title of the reference should be changed.

Reviewer #2: Comments to PONE-D-24-55825R1

I would like to express my sincere gratitude for being appointed as a reviewer for PONE-D-24-55825R1. This paper addresses a critical issue regarding the optimization of lymph node dissection in pancreatic cancer surgery, and I recognize its significant clinical relevance.

Based on the PBP responses, the authors addressed points raised by reviewers properly.

**Do you want your identity to be public for this peer review?** For information about this choice, including consent withdrawal, please see our Privacy Policy

Reviewer #1: No

Reviewer #2: **Yes: ** Yasutoshi Kimura

---

## [Author Response · Author response to Decision Letter 2]

12 Mar 2025

Response to the comments of the editor and the reviewers

Editor

Thank you for reviewing and handling our manuscript entitled " A multicenter prospective study to determine the optimal range of lymph node dissection in pancreatic cancer surgery after neoadjuvant chemotherapy (LYMRIN-Trial): Project study by the Japan Pancreas Society and JON 2302-P". We re-wrote the manuscript and table according to reviewers’ comments. We also responded to the Journal requirements one by one. The manuscript was brushed up. We have marked in red the changes made in the manuscript. We added names of collaborators on the title page.

Journal Requirements:

[Please review your reference list to ensure that it is complete and correct. If you have cited papers that have been retracted, please include the rationale for doing so in the manuscript text, or remove these references and replace them with relevant current references. Any changes to the reference list should be mentioned in the rebuttal letter that accompanies your revised manuscript. If you need to cite a retracted article, indicate the article’s retracted status in the References list and also include a citation and full reference for the retraction notice.]

We reviewed the reference list to ensure that it is complete and correct.

Reviewer #1

Thank you very much for your peer review.

[#1. In the response, the authors noted “However, this study includes several requirements with additional interventions. In the pancreas head tumor, No. 14 op lymph node (Lymph nodes along the superior mesenteric artery; opposite side of tumor) is not a regional lymph node in Japanese classification of pancreatic carcinoma by the Japan Pancreas Society: Eighth edition, but our protocol needs to dissect No. 14op lymph nodes with pancreatoduodenectomy.” If the protocol required additional interventions, they should be clearly stated in “Treatment protocol”. In addition, Table 2 included No. 14op in regional lymph nodes of the tumor in the pancreas head. As the authors noted No. 14op is not regional lymph nodes.]

We appreciate the professional comments. As the reviewer said, if additional interventions are required, we should describe the comment in treatment protocol. We added the comments below. In addition, we changed the title of Table 2 from “Regional lymph nodes by tumor location” to “Lymph nodes to be dissected by tumor location”.

This study includes several requirements with additional interventions. In the pancreas head tumor, No. 14 op lymph node (Lymph nodes along the superior mesenteric artery; opposite side of tumor) is not a regional lymph node in General rules for the study of pancreatic cancer, 8th edition [10], but our protocol needs to dissect No. 14op lymph nodes with pancreatoduodenectomy.

(Page 13, line 236)

Pancreatoduodenectomy for pancreatic head tumors requires dissection of 14op lymph node, which is outside the regional lymph node.

(Page 14, line 258)

[Line 238-240, the title of the reference should be changed.]

We appreciated the comments. We changed the title of the reference as follows:

Japanese classification of pancreatic carcinoma by the Japan Pancreas Society: Eighth edition

(Page 13, Line 255)

Reviewer #2

Thank you very much for your peer review.

---

## [Decision Letter · Decision Letter 2]

May 23 2025

Dear Dr. Fujii,

Thank you for submitting your manuscript to PLOS ONE. After careful consideration, we feel that it has merit but does not fully meet PLOS ONE’s publication criteria as it currently stands. Therefore, we invite you to submit a revised version of the manuscript that addresses the points raised during the review process.

We look forward to receiving your revised manuscript.

Kind regards,

Peng Zhang, Ph.D.

Academic Editor

PLOS ONE

Journal Requirements:

Reviewers' comments:

Reviewer's Responses to Questions

**Comments to the Author**

1. Does the manuscript provide a valid rationale for the proposed study, with clearly identified and justified research questions?

Reviewer #1: Yes

2. Is the protocol technically sound and planned in a manner that will lead to a meaningful outcome and allow testing the stated hypotheses?

Reviewer #1: Yes

3. Is the methodology feasible and described in sufficient detail to allow the work to be replicable?

Reviewer #1: Yes

4. Have the authors described where all data underlying the findings will be made available when the study is complete?

Reviewer #1: Yes

5. Is the manuscript presented in an intelligible fashion and written in standard English?

Reviewer #1: Yes

You may also provide optional suggestions and comments to authors that they might find helpful in planning their study.

Reviewer #1: Thank you for giving opportunity to review this revised manuscript. The issue I previously pointed out has been revised properly.

**Do you want your identity to be public for this peer review?** For information about this choice, including consent withdrawal, please see our Privacy Policy

Reviewer #1: No

---

## [Author Response · Author response to Decision Letter 3]

2 May 2025

Response to the comments of the editor and the reviewers

Editor

Thank you for reviewing and handling our manuscript entitled " A multicenter prospective study to determine the optimal range of lymph node dissection in pancreatic cancer surgery after neoadjuvant chemotherapy (LYMRIN-Trial): Project study by the Japan Pancreas Society and JON 2302-P". We re-wrote the manuscript and reference according to Journal requirements. The manuscript was brushed up. We have marked in red the changes made in the manuscript.

Journal Requirements:

[Please review your reference list to ensure that it is complete and correct. If you have cited papers that have been retracted, please include the rationale for doing so in the manuscript text, or remove these references and replace them with relevant current references. Any changes to the reference list should be mentioned in the rebuttal letter that accompanies your revised manuscript. If you need to cite a retracted article, indicate the article’s retracted status in the References list and also include a citation and full reference for the retraction notice.]

We reviewed the reference list and made changes to several references.

Citation numbers 2 and 3 were changed to the journal articles because they were citing online sites.

2. Chen C, Yin L, Lu C, Wang G, Li Z, Sun F, et al. Trends in 5-year cancer survival disparities by race and ethnicity in the US between 2002-2006 and 2015-2019. Sci Rep. 2024;14(1):22715. doi: 10.1038/s41598-024-73617-z.

3. Isaji S, Mizuno S, Windsor JA, Bassi C, Fernández-Del Castillo C, Hackert T, et al. International consensus on definition and criteria of borderline resectable pancreatic ductal adenocarcinoma 2017. Pancreatology. 2018;18(1):2-11. doi: 10.1016/j.pan.2017.11.011.

(Page 24)

Citation number 4 was changed because the journal article was published online ahead.

4. Unno M, Motoi F, Matsuyama Y, Satoi S, Toyama H, Matsumoto I, et al. Neoadjuvant chemotherapy with gemcitabine and S-1 versus upfront surgery for resectable pancreatic cancer: Results of the randomized phase II/III Prep-02/JSAP05 trial. Ann surg. 2025; online ahead of print. doi: 10.1097/SLA.0000000000006730.

(Page 24)

Citation 12 was removed because it was non-public data using the National clinical database.

Reviewer #1

Thank you very much for your peer review.

---

## [Editor Report · Decision Letter 3]

A multicenter prospective study to determine the optimal range of lymph node dissection in pancreatic cancer surgery after neoadjuvant chemotherapy (LYMRIN-Trial): Project study by the Japan Pancreas Society and JON 2302-P

PONE-D-24-55825R3

Dear Dr. Fujii,

We’re pleased to inform you that your manuscript has been judged scientifically suitable for publication and will be formally accepted for publication once it meets all outstanding technical requirements.

Kind regards,

Peng Zhang, Ph.D.

Academic Editor

PLOS ONE
---

## [Editor Report · Acceptance letter]

PONE-D-24-55825R3

PLOS ONE

Dear Dr. Fujii,

I'm pleased to inform you that your manuscript has been deemed suitable for publication in PLOS ONE. Congratulations! Your manuscript is now being handed over to our production team.

Kind regards,

on behalf of

Prof. Peng Zhang

Academic Editor

PLOS ONE